# Brown Seaweed Extract (BSE) Application Influences Auxin- and ABA-Related Gene Expression, Root Development, and Sugar Yield in *Beta vulgaris* L.

**DOI:** 10.3390/plants12040843

**Published:** 2023-02-13

**Authors:** Giovanni Bertoldo, Claudia Chiodi, Maria Cristina Della Lucia, Matteo Borella, Samathmika Ravi, Andrea Baglieri, Piergiorgio Lucenti, Bhargava Krishna Ganasula, Chandana Mulagala, Andrea Squartini, Giuseppe Concheri, Francesco Magro, Giovanni Campagna, Piergiorgio Stevanato, Serenella Nardi

**Affiliations:** 1Department of Agronomy, Food, Natural Resources, Animals and Environment (DAFNAE), Campus of Agripolis, University of Padova, Viale dell’Università 16, 35020 Legnaro, Italy; 2Crop Production and Biostimulation Laboratory, Interfacultary School of Bioengineers, Université Libre de Bruxelles, Campus Plaine CP 245, Bd du Triomphe, 1050 Brussels, Belgium; 3Dipartimento di Agricoltura Alimentazione e Ambiente (Di3A), Università di Catania, Via S. Sofia 100, 95123 Catania, Italy; 4Sofbey SA, Cso S. Gottardo 34, 6830 Chiasso, Switzerland; 5CO.PRO.B—Cooperativa Produttori Bieticoli, Via Mora 56, 40061 Minerbio, Italy

**Keywords:** sugar beet, root traits, RNA-seq, RT-qPCR, molecular pathways, sugar yield

## Abstract

The molecular and phenotypic effects of a brown seaweed extract (BSE) were assessed in sugar beet (*Beta vulgaris* L.). Transcript levels of BSE-treated and untreated plants were studied by RNA-seq and validated by quantitative real-time PCR analysis (RT-qPCR). Root morphology, sugar yield, and processing quality traits were also analyzed to better elucidate the treatment effects. RNA-seq revealed 1019 differentially expressed genes (DEGs) between the BSE-treated and untreated plants. An adjusted *p*-value < 0.1 and an absolute value of log2 (fold change) greater than one was used as criteria to select the DEGs. Gene ontology (GO) identified hormone pathways as an enriched biological process. Six DEGs involved in auxin and ABA pathways were validated using RT-qPCR. The phenotypic characterization indicated that BSE treatment led to a significant increase (*p* < 0.05) in total root length and the length of fine roots of plants grown under hydroponics conditions. The sugar yield of plants grown under field conditions was higher (*p* < 0.05) in the treated field plots compared with the control treatment, without impacting the processing quality. Our study unveiled the relevant effects of BSE application in regulating auxin- and ABA-related gene expression and critical traits related to sugar beet development and yield.

## 1. Introduction

Environmental concerns about climate change and food security are prompting modern agriculture to find environmentally friendly ways to sustain crop productivity and reduce its reliance on chemical fertilizers [1]. Plant biostimulants (PBs) have evolved into unique and sustainable agricultural inputs during the last few decades [2]. Their impact on plants depends on various mechanisms, including the ability to promote hormone-like activity and to stimulate plant−soil microbe interactions [3,4,5,6]. Seaweed extracts (SE) are a major class of plant biostimulants subjected to the study of multidimensional plant responses. Seaweeds are macroscopic algae that can be found in coastal and aquatic habitats. They are characterized by high quantities of polysaccharides, polyunsaturated fatty acids (PUFAs), enzymes, and bioactive peptides [7,8,9,10,11].

Treatments with seaweed extracts can impact the transcription levels of plant genes involved in the synthesis of growth hormones, such as auxin, cytokinin, and abscisic acid (ABA) [12,13,14,15,16,17,18]. The already demonstrated impact of SE on plant hormone activities has, as a consequence, had multifaceted effects on plant development and yield, but the mechanism that regulates such plant responses has yet to be understood and needs careful assessment.

Sugar beet is a root crop that grows mainly in temperate areas and accounts for approximately 20% of the global yearly sugar output [19]. The use of PBs during critical stages could hasten root growth, allowing for rapid soil colonization and, consequently, more efficient water and nutrient uptake. Barone et al. [20] identified extracts from microalgae such as *Scenedesmus quadricauda* and *Chlorella vulgaris* as potential biostimulants in the initial growth stages of sugar beet cultivation, enhancing nitrogen uptake and root development.

The combination of high-throughput transcriptomics and phenomics was the most effective approach to characterize the PBs effects and their mode of action on important crops such as corn, soybean, and tomato [21,22]. Remarkably, the RNA-seq approach is used not only to clarify the mode of action of PBs, but also as a successful strategy for developing new products [23]. In this work, we aimed to assess the effects of a brown seaweed extract (BSE) in sugar beet grown under hydroponics and open field. RNA-seq and quantitative real-time PCR (RT-qPCR) approaches substantiated the molecular impacts of BSE. The root morphology, sugar yield, and quality traits were also analyzed to better elucidate the treatment effects.

## 2. Results

### 2.1. Chemical Characterization of the BSE Extract

The constituents of BSE biostimulants were analyzed by determining the amounts of specific major components such as dry matter, ash, and total carbohydrates and polyphenols (Table 1). Carbohydrates and phenolic compounds accounted for up to 60% and 15% of all matter, respectively.

### 2.2. Transcriptomic Analysis of Plants Grown in the Open Field

After removing the low-quality adaptor and barcode sequences, 91.29, 88.51, and 90.82 total million raw reads were obtained from the transcriptome libraries of samples treated with 1 mL L^−1^, 2 mL L^−1^, and 4 mL L^−1^ concentrations, respectively. The average total number of reads for each sample was 6,495,945, with an average overall alignment rate of 78.41%. The mapped reads ensued in the recognition of a total of 24,206 genes in all of the samples that were used for further analysis on the effect induced by BSE leaf treatment in the Beta vulgaris L. transcriptome.

#### 2.2.1. Identification of DEGs Response to BSE Treatment

DEGs were analyzed for the three different BSE concentrations (1, 2, and 4 mL L^−1^), at three time points (24, 48, and 72 h after the treatment), at phenological stage BBCH32. A total of 813 significant DEGs (adj-*p* < 0.1) were found 24 h after the treatment, irrespective of the dilution used. Among them, 641 were upregulated and 172 were downregulated. A total of 560 significant DEGs (adj-*p* < 0.1) were found 48 h after the treatment, irrespective of the dilution. Among them, 133 were upregulated and 427 were downregulated. The results for 72 h were not reported as the DEGs analysis did not show any significant differentially expressed genes.

The 2 mL L^−1^ concentration was the one stimulating the highest number of upregulated genes 24 h after the treatment (326 DEGs) and at 48 h (57 DEGs). In addition, the highest number of downregulated genes 48 h after the treatment were obtained in the samples treated with the same BSE dose.

With the BSE dose equal to 1 mL L^−1^, the highest number of downregulated DEGs at 24 h (111 DEGs at 24 h) were found. This dilution was also the one with the less upregulated DEGs in both of the timings (154 DEGs at 24 h and 24 at 48 h after treatment) (Appendix A). 

Some genes have been found to be differentially expressed in response to the three dilutions (Figure 1: 24 h after treatment, most of the genes were upregulated from the treatment (Figure 1a), while 48 h after the treatment, most of the genes were downregulated from the treatment (Figure 1b). The 2 mL L^−1^ dilution seemed to be the one modulating the highest number of genes: 266 specific upregulated genes after 24 h and 72 specific downregulated genes after 48 h. These results underline a dose-dependent effect of BSE on the gene expression and the time-dependent effect of the BSE treatment.

Finally, the up- (Appendix A) and down- (Appendix A) regulated genes after 24 h and 48 h treatments were sorted based on their enriched functional category (biological process). 

A cluster analysis on enriched GO terms highlighted specific pathways homogeneous in their DEGs composition. A consistent part of the upregulated genes was clustered in the hormonal pathways. “Cellular response to auxin stimulus” was the category with the highest fold enrichment value. Among the downregulated genes, the GO category with the highest number of DEGs was the “organonitrogen compound metabolic process”.

#### 2.2.2. Candidate Genes Validation

The expression of six DEGs (*ARF19*, *NH23*, *YUCCA6*, *PYL4*, *MYB30*, and *PP2C62*) involved in hormonal regulation was evaluated using RT-qPCR in leaf samples collected from plants grown in pots and treated with BSE leaf treatment. Detailed statistical results are shown in Appendix A.

Overall *ARF19*, *NH23,* and *YUCCA6* are part of the AUXIN pathway, while *PP2C62*, *PYL4*, and *MYB30* belong to the ABA pathway. Comparative expression levels of the six identified genes indicated distinct expression levels between the BSE-treated and untreated plants.

*ARF19*, *NH23*, and *YUCCA6* were upregulated at both time points in response to the treatments (Figure 2). *PYL4*, *MYB30*, *MYB30,* and *PYL4* were downregulated, while *PP2C62* was upregulated in the treated plants but downregulated in the untreated control (Figure 3). 

### 2.3. Root Morphological Analysis of Plants Grown in Hydroponics

The total root length and length of the fine roots were measured to evaluate the effect of the hydroponic BSE treatment on the root system. Morphological changes were observed after 48 h of treatment (Figure 4). Significant variations (*p* < 0.05) with respect to the untreated plants were observed in the total root length and fine roots length with 0.2 mL L^−1^ and 0.1 mL L^−1^ dilutions (Figure 4). Only the higher concentration (2 mL L^−1^) showed a detrimental effect on the root length.

### 2.4. Yield Analysis

The effects of the foliar application of the BSE biostimulant on sugar beet in open field conditions were measured by evaluating the yield and other quality parameters on sugar beet taproot. Plants treated with BSE presented a higher root yield and sugar yield (Table 2). No relevant effect of treatment on sugar purity (K, Na, and α-amino-N constituents) was found in the sugar beet root juice in the treated plants compared with the untreated plants.

## 3. Discussion

Plant biostimulants are a new group of agricultural products that boost crop quality and yield while guarding against biotic and abiotic challenges [24]. Effects vary depending on application and dose and are also affected by various agronomical and environmental factors [25]. The ability to predict plant response to biostimulants is necessary for the outgrowth of sustainable agriculture. The impact of the BSE biostimulant on plant growth is multidimensional and leads to increased production [26]. They, instead, operate ambiguously on plant metabolism by generating signaling cascades triggering reactions that lead to biotic and abiotic stress mitigation together with an increased growth and productive performance as a result [27,28]. Because of the vast range of bioactive components found in seaweeds, there is little knowledge about the intricate mechanisms through which seaweed extracts affect plant development processes [29,30]. 

In this study, we aimed to assess the effects of a brown seaweed extract (BSE) on sugar beet grown in both open fields and hydroponics. First, the chemical characterization of the BSE extract revealed that carbohydrates are a major component of the product. Seaweed-derived carbohydrates can trigger plant biostimulation, as reported by Carmody et al. [31]. The carbohydrates in biostimulants may act by changing plant signaling cascades to activate defense reactions in response to abiotic and biotic stimuli. In addition, carbohydrates may act as a source of energy and carbon for endophytic and non-endophytic microbial populations [32]. 

After the chemical characterization, we focused on the plant’s transcriptome because biostimulants have a broad spectrum of activity encompassing numerous plant metabolic pathways and biological processes, mainly in hormone regulation. Transcriptomics allowed us to identify a set of BSE-responsive DEGs. In particular, the detailed characterization of the expression levels of six DEGs (*ARF19*, *NH23*, *YUCCA6*, *PYL4*, *MYB30*, and *PP2C62*) involved in hormonal regulation was evaluated using RT-qPCR in leaf samples treated with BSE. 

*ARF19, NH23,* and *YUCCA6* are part of the AUXIN pathway [33,34,35]. BSE treatment upregulated the transcription levels of these genes. The *ARF19* gene belongs to the *ARF* gene family that controls many plants’ developmental phases through the proteins involved in DNA binding, transcriptional activation, or repression. Upregulation of *ARF19* controls the auxin-responsive gene expression related to the lateral root initiation in *Arabidopsis thaliana* [36]. The *NH23* gene belongs to the Nudix family and plays an active part in cellular homeostasis and plant signaling, affecting several outputs in such as hormone signaling and pathogen defense [34]. The *YUCCA6* gene belonging to the *YUCCA* family has a crucial role as the primary endogenous auxin biosynthesis pathway that is involved in major biological processes mediated by the activity of auxin [37]. Overexpression of the *YUCCA6* gene leads to elevated auxin levels and the induction of auxin-responsive genes together with a large increase in inflorescence height and altered leaf morphology in *Arabidopsis thaliana* [38]. 

Out of six selected genes, *PYL4*, *MYB30*, and *PP2C62* are part of the ABA pathway [39,40,41]. In the treated plants, *PYL4* and *MYB30* were downregulated. *PYL4* is known to encode the ABA receptor. Dittrich et al. [39] reported that *PYL4*/*5* is needed for a CO_2_-induced guard-cell response and for regulating the stomatal functioning in *Arabidopsis thaliana*. If upregulated, *PYL4* acts as an inductor of stomatal closure through the regulation of the turgor of the guard cells. The transcript overaccumulation of this gene can trigger the degenerative process of leaf senescence. *MYB30* is described as a transcription factor acting as an ABA-responsive factor. It has been reported to play an important role in root elongation through ROS-dependent processes in the ABA signaling pathways [40]. Moreover, the *R2R3-MYB* transcription factor *MYB30* has been identified as an effective regulator of the hypersensitive response (HR) programmed cell death linked to pathogen resistance in plants and brassinosteroid (BR) signaling [42]. *MYB30* enhances HR and BR signaling by instantly interacting with *BRI1-EMS-SUPPRESSOR 1(BES1)* and improving its activity [43]. Furthermore, *MYB30* influences HR and disease resistance by regulating the salicylic acid (SA) status and expressing SA-related genes [44]. *PP2C62* was upregulated in the treated plants. *PP2C62* is a negative regulator of ABA synthesis, and its upregulation can lead to a reduction in the ABA content, as well as in a general non-stressed status, due to the repressed ABA sensing mechanisms, which can orchestrate a generalized improved photosynthetic machinery, because of the unaltered stomatal conductance in response to low ABA sensing [45,46,47,48,49]. The *PYR*/*PYL*s, *PP2C*, along with *SnRK2*s and ABF ABA receptors form the core network of ABA signal control [41]. It is well established that the gene family PP2Cs are at the center of the ABA signaling network, and the upregulation of these genes inhibits the ABA receptors system and blocks the downstream ABA-responsive gene. 

In the present work, root morphological features such as the total root length and length of the fine roots in hydroponics significantly increased following BSE application. These findings are especially intriguing for sugar beet as the above root traits influence plant development and water−nutrient uptake, which improves the final sugar yield [50]. Our results showed, in fact, that the treated plants had a relatively high sugar yield when compared with the untreated plants. We found no discernible variations in the impurity level between the treated and control plants. The upregulation of auxin-related genes observed in the treated plants could be as a result of an auxin-driven response at a phenotypical level, improving the root development. Auxin engages in a comprehensive growth activity, including cell division in the root pericycle, which is essential for the beginning and elongation of lateral roots [51]. 

The schematic representation of the BSE mode of action influencing the auxin- and ABA-related gene expression is summarized and given in Figure 5.

## 4. Materials and Methods

### 4.1. Chemical Characterization of the BSE Extract

The BSE extract was provided by Sofbey SA (Chiasso, Switzerland) and was extracted from brown seaweed *Ascophyllum nodosum.* Dry weight (DW) was measured by placing the samples of BSE in a drying oven at 105 °C until a standard weight was reached. The samples were cooled for two hours inside a closed bell jar, and then the obtained dry matter was weighed again. The BSE ash content was estimated by incineration of the sample in a muffle furnace at 550 °C to constant mass, and was expressed as % with respect to DW. The carbohydrate content was assessed according to Moxley and Zhang [52] by ion exchange chromatography using a Dionex DX500 system equipped with CarboPac PA20 using an isocratic elution of 20 mM NaOH. The sensor (pulsed amperometric-EDet1) used the Gold Standard PAD waveform with an AgCl reference electrode, which included the following electrode potentials set as waveform A: E1: +0.1 V for 400 ms. E2: −2.0 V for 1 ms. E3: +0.6 V for 1 ms. E4: −0.1 V for 6 ms. The samples were prepared by treating 100 mg of the dried BSE with 3 mL of 72% H2SO4 (*w*/*w*) at 30 °C for 20 min, then diluted with 84 mL of distilled water and 4% H2SO4 (*w*/*w*), and finally autoclaved at 121 °C for 20 min [53]. The total carbohydrate content was expressed as g kg^−1^ of the extract dry weight (DW). Lipids were extracted from 250 mg of freeze-dried samples using dichloromethane, following the method described by Folch et al. [54]. The total lipid content was expressed as g kg^−1^ of the dry weight of the extract. The complete protein content was quantified according to the Bradford method [55] using BSA as a standard curve and was expressed as mg protein g kg^−1^ of DW of extract. The total phenols were measured according to Chatris et al. [56]. Soluble phenolic acids were extracted with 3 mL pure methanol (1:10 *w*/*v*). The extracts were maintained in an ice bath for 30 min and then centrifuged at 5000× *g* for 30 min at 4 °C. The supernatants were stored at −20 °C until analysis.

### 4.2. Plant Material and Growing Conditions

A diploid sugar beet hybrid (Cv. Beniamina, KWS, Einbeck, Germany) was used for this study. This hybrid is tolerant to *Cercospora beticola* Sacc. and *Heterodera schachtii*.

#### 4.2.1. Field Experiment

Field trials were carried out between March and August 2020 and 2021 in San Martino di Venezze, Rovigo, Italy (45°06′12.9″ N. 11°53′52.5″ E). The experimental design comprised four randomized blocks. Each randomized block was split into four sub-plots of 2.7 × 10 m. The sugar beet plant density in each sub-plot was 10 plants m^−2^. An additional control pot was added to the experiment. The foliar biostimulant treatment was applied at BBCH32 and BBCH40. The plants were treated with foliar sprays of BSE solutions at different dilutions (4 mL L^−1^, 2 mL L^−1^, and 1 mL L^−1^). The leaf samples were collected before treatments and 24 h, 48 h, and 72 h after treatment. Then, the samples were immediately transferred to dry ice and stored at −80 °C for the transcriptome analysis.

#### 4.2.2. Pots Experiment

Sugar beet seedlings were grown in 13 cm diameter pots filled with standard peat substrate, with pH 7. A mineral-based slow-release fertilizer (nitrophoska) was applied before seedling germination to the substrate at a rate of 20 g per pot. The fertilizer chemical composition included nitrogen, phosphorus, and potassium at concentrations of 12%, 12%, and 17%, respectively; the fertilizer also had an additional 2% of MgO, 24% of sulfur, 0.02% of boron, and 0.10% of zinc. Each pot was irrigated with around 250 mL of water every two days. Water in excess was resupplied until complete absorption by the peat substrate. Pots were maintained for 50 days in a climatic chamber at 25/20 °C and a 16/8 light/dark photoperiod. The foliar biostimulant treatment was applied at BBCH32. Plants were treated with foliar sprays of different BSE dilutions (2 mL L^−1^, 1 mL L^−1^, 0.5 mL L^−1^, 0.2 mL L^−1^, and 0.1 mL L^−1^). The leaf samples were collected before treatments and 24 h and 48 h after treatments. Sampling was performed by taking two leaf disks per plant from each experimental condition. A total of four biological replicates were collected from the treated and untreated plants. Then, the samples were immediately stored at −80 °C for the validation analysis.

#### 4.2.3. Hydroponics Experiment

Sugar beet seedlings were grown under hydroponics to determine root morphological traits. The seeds were sterilized for 5 min in 76% ethanol and rinsed in distilled water three times. The seeds were put on wet filter paper and incubated in a growth chamber at 25 °C for 48 h. Germinated seeds were transplanted into 500 mL glass pots with a Hoagland solution [57] (Arnon and Hoagland, 1940). After eight days, 30 plants of each replicate were treated with multiple BSE dilutions (2 mL L^−1^, 1 mL L^−1^, 0.5 mL L^−1^, 0.2 mL L^−1^, and 0.1 mL L^−1^). To select the appropriate BSE doses, preliminary tests were conducted. The samples were collected 48 h after BSE application. The experiment was performed in triplicate.

### 4.3. RNA Sequencing and Differential Gene Expression Analysis

RNA sequencing was carried out in leaf samples collected from sugar beet grown under field conditions. Samples were collected at 24 h, 48 h, and 72 h after treatment. Sampling was performed using two leaf disks from four plants from each experimental condition. The RNA sequencing protocol was performed entirely in-house and has also been described in Della Lucia et al. [21]: mRNA was extracted using the Dynabeads mRNA Direct Micro Kit (Thermo Fisher Scientific, Carlsbad, CA, USA), then quantified using an Agilent 4150 TapeStation system (Agilent Technologies, Santa Clara, CA, USA). Sequencing libraries were prepared using Ion Total RNA-Seq Kit v2 (Thermo Fisher Scientific). Libraries were quantified through D1000 screen tape (Agilent Tapestation 1500), normalized to obtain a molar concentration of 100 pM, and then pooled and sequenced using three Ion 540™ Chips on the Ion Torrent S5 System (Thermo Fisher Scientific). In all of the steps, the manufacturer’s instructions were followed. Low-quality reads were removed from the raw RNA-seq data with a phred-like Q value > 20. Screened reads were mapped to the reference sugar beet genome (publicly accessible from NCBI, GenBank accession GCA 000188115.3) by using Bowtie2 (v2.4.2) [58]. Samtools (v1.11) [59] was used to examine the mapped files, and raw read values for all of the annotated genes were determined using bedtools multiBamCov v2.30.0 [60]. Non-informative data were removed by filtering the genes with a total expression level less than 20 reads. To execute the inferential analysis and identify the differentially expressed genes (DEGs) among the different experiments, the DESeq2 R package (v.1.30.0) [61] was used. A *p*-value < 0.05 and a |log2-fold change| ≥ 1.0 were used as the criteria of significance to select the DEGs. 

### 4.4. Validation of Selected DEGs by RT-qPCR 

The validation of the selected DEGs by RT-qPCR was carried out in leaf samples collected from sugar beet plants grown in pots. The total RNA for validation was extracted using the RNeasy plant mini kit (Qiagen, Hilden, Germany) following the manufacturer’s instructions. Primer Express V3.0 from Thermo Fisher Scientific was used to build primers using mRNA sequences selected from the reference sugar beet genome (publicly accessible from NCBI, GenBank accession GCA 000188115.3). The primer sequences are reported in Table 3.

Real-time quantitative reverse transcription PCR (RT-qPCR) amplification and detection were performed using a Quant Studio 12K Flex Real-Time PCR (Thermo Fisher Scientific) and the Quantitect SYBR Green RT-PCR kit (Qiagen) using the one-step protocol. The 10 μL reaction mix was composed of 5 μL of Quantitect SYBR Green master mix, 0.5 μL of retro-transcriptase, 0.5 μL of both forward and reverse primers, 2.5 μL of nuclease-free water, and 1 μL of RNA. Each sample was run in triplicate. The thermocycling conditions were: 15 min–95 °C, 45 cycles of 30 s–95 °C, 30 s–55 °C, and 30 s–72 °C.

The resulting threshold cycle (Ct) values were standardized against the mean transcript levels of three housekeeping genes (*GAPDH*, *Actin*, and *UBI*) using the ΔΔCt method, where Ct is calculated as the difference between the target gene’s Ct and the control gene’s Ct [62,63]. Then, to assess the fold change, the level of expression before the treatment was used as a control, for the treated and untreated conditions, at different timings.

### 4.5. Root Morphological Analysis

Root morphological traits, such as the total root length and fine root length, were evaluated on seedlings grown in hydroponics using a scanner-based image processing method (WINRHIZO Pro Regent Instruments, Quebec City, QC, Canada) after 48 h of BSE treatment. To improve contrast, the root systems were stained for 15 min with 0.1% (*w*/*w*) toluidine blue (Sigma-Aldrich, Montréal, QC, Canada). The root systems were kept in 3 mm of water in a Plexiglas tray, and the lateral roots were distributed to reduce root overlap. The tray was scanned (STD-1600 EPSON) at a resolution of 1200 dpi. The total root length and fine root length were measured using the WINRHIZO software.

### 4.6. Yield Measurements

Sugar beet yield traits such as root yield, sugar yield, and processing quality-related parameters were determined in the field experiments. A total of 100 sugar beet roots were collected at BBCH 49 from each subplot. The roots of each plant were cleaned before being sawed into 1 kg of micronized tissues (brei) using a specialized saw (AMA-KWS. AMA Werk GmbH, Alfeld, Germany). Approximately 70 g of homogenized brei samples were promptly frozen at −40 °C. The sugar content and major non-sugars were determined using an automatic brei mixer following cold digestion of the brei in lead acetate 0.75% (*w*/*w*) solution [64] (Venema Automation b.v. Groningen, The Netherlands). A Thorn-Bendix 243 polarimeter (Bendix Corp, Nottingham, UK) was utilized to quantify the sugar content. Finally, a flame photometer was employed to determine the K and Na contents (Model IL 754. Instrumentation Laboratory S.p.A., Milan, Italy). The colorimetric analysis (PM2K; Carl Zeiss GmbH, Oberkochen, Germany) method proposed by Kubadinow and Wieninger [65], and Stevanato et al. [66] was employed to measure the α-amino N. The purity was calculated as the proportion of sugar from the roots that the manufacturer could extract and was quantified at 405/492 nm using the plate reader Uniplan AIFR-01 (CJSC Picon, Moscow, Russia) [65,66].

## 5. Statistical Analysis

The statistical analysis was performed using Statistica 13.0 (StatSoft, Tulsa, OK, USA) and Sigma Plot 14.0 (Systat Software, Palo Alto, CA, USA) packages. The data are expressed as mean values ± standard errors. One-way ANOVA analysis was carried out to determine whether untreated and treated samples differed in terms of the evaluated variables. In the case of significant difference (*p* value < 0.05), the means were separated using Duncan’s method.

## 6. Conclusions

In conclusion, brown seaweed extract (BSE) application influences auxin- and ABA-related gene expression with improvements in the root morphological traits and sugar yield, without impacting the processing quality. Particularly, the root morphological analysis of the treated plants highlighted that the BSE stimulatory activity led to better root morphological development, allowing for rapid soil colonization and, consequently, more efficient water and nutrient uptake. These critical traits correlate with the remarkable yield attributes in field-grown sugar beets. The application of the BSE extract at the field level could enhance the ability of sugar beet to cope with environmental stressors. Further study should also investigate this trend in other crops so as to search for shared or unique responses to BSE and a common ground for the BSE mode of action.

## Figures and Tables

**Figure 1 plants-12-00843-f001:**
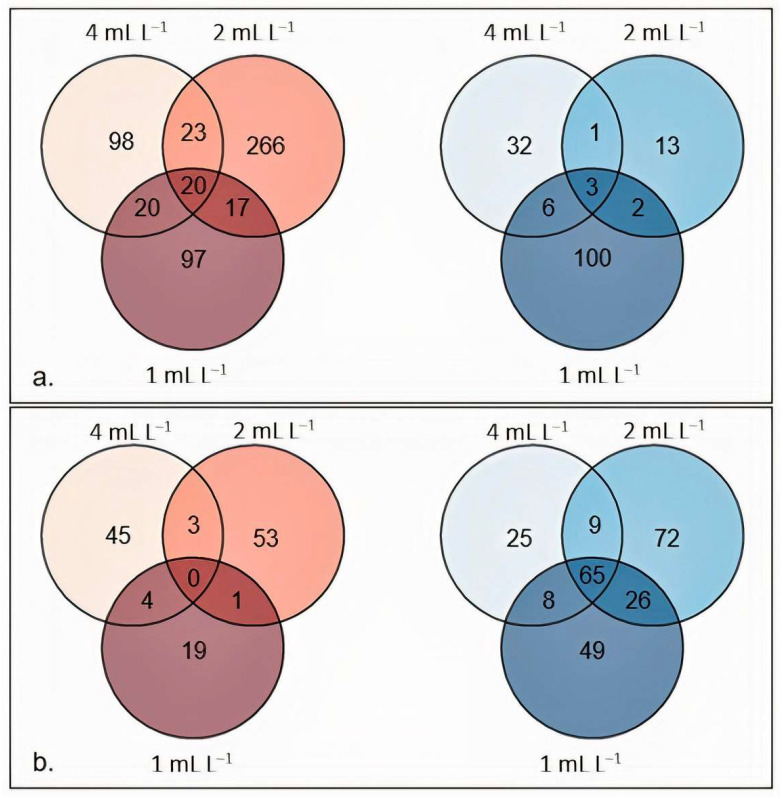
Number of DEGs specific and unique for each dilution. Upregulated (red) and downregulated (blue) genes at the three BSE concentrations with respect to the control 24 h after treatment (**a**) and 48 h after treatment (**b**).

**Figure 2 plants-12-00843-f002:**
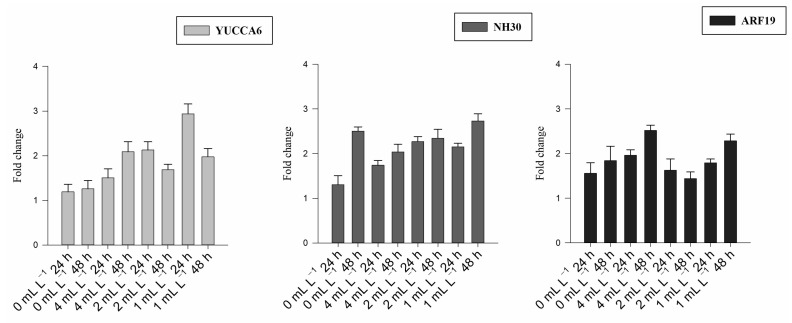
Fold change for the gene expression analysis of the auxin-related genes 24 and 48 h after treatment for all the BSE concentrations (0 mL L^−1^: untreated; 4 mL L^−1^; 2 mL L^−1^; 1 mL L^−1^). Each bar-plot shows the mean of eight replicates with the standard error of the mean.

**Figure 3 plants-12-00843-f003:**
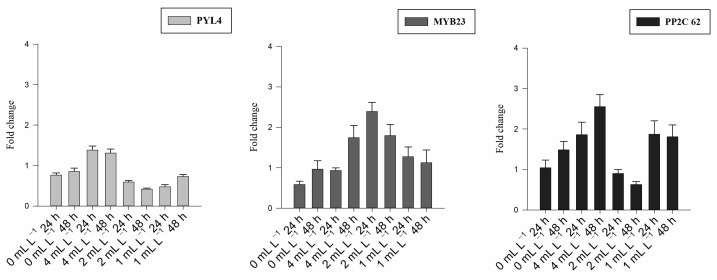
Fold change for the gene expression analysis of the ABA-related genes 24 and 48 h after treatment for all the BSE concentrations (0 mL L^−1^: untreated; 4 mL L^−1^; 2 mL L^−1^; 1 mL L^−1^). Each bar-plot shows the mean of eight replicates with the standard error of the mean.

**Figure 4 plants-12-00843-f004:**
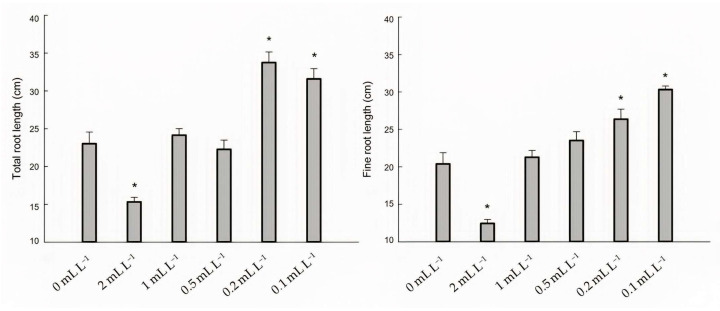
Total root length (cm) and fine root length (cm) of seedlings grown in hydroponics 48 h after treatment. “*” identifies the group of treated plants whose means are significantly different from the untreated (0 mL L^−1^) at *p* < 0.05, after Duncan’s post hoc test.

**Figure 5 plants-12-00843-f005:**
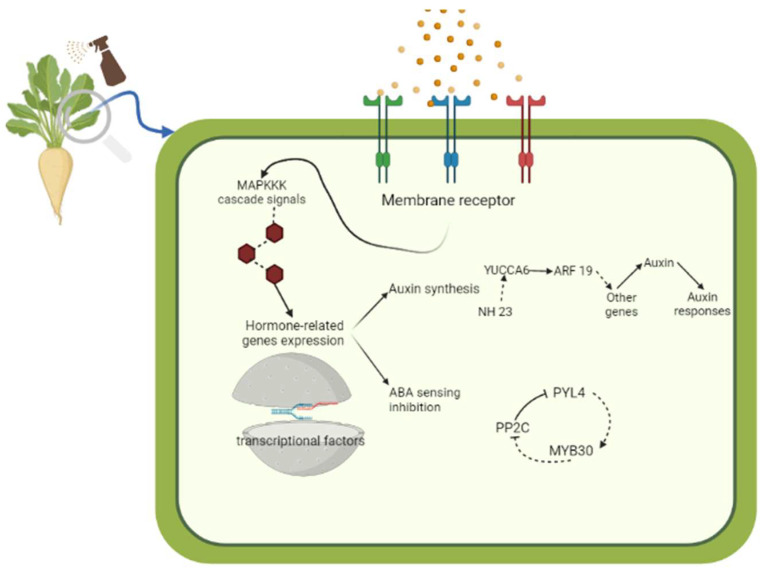
Putative representation of the BSE mode of action. The BSE foliar treatment is perceived by a complex sensing mechanism made of different membrane receptors that can trigger a classic MAPKKK cascade signaling, resulting in a regulatory response of different pathways. In this study, it is shown that the different dosages of BSE treatment can alter the expression level of the key genes involved in auxin- and ABA-related gene expression with consequent effects on the root development and sugar yield in the sugar beet.

**Table 1 plants-12-00843-t001:** Composition of the BSE biostimulant. Each value is expressed based on the total dry matter of the BSE extract.

Composition	
Dry matter (g L^−1^)	90.39 ± 2.3
Ash (%)	29.58 ± 0.9
Carbohydrate (g kg^−1^)	387.70 ± 12.1
Lipids (g kg^−1^)	2.50 ± 0.1
Protein (g kg^−1^)	37.80 ± 0.8
Total phenolic compounds (g kg^−1^)	101.20 ± 1.8

**Table 2 plants-12-00843-t002:** Mean values of sugar yield, root yield, and processing quality traits in the BSE-treated and untreated sugar beet. The results are expressed as the mean of four randomized replicates with 60 plants each. ANOVA was used to evaluate the differences between the treatments with a 0.05 *p*-value threshold. Mean values followed by asterisk differ significantly from the untreated samples.

Treatment	Root Yield (t ha^−1^)	Sugar Yield (t ha^−1^)	Potassium (meq % °S)	Sodium (meq % °S)	α-Amino N (meq % °S)	Sugar Purity (%)
Control	75.6	11.4	26.12	5.56	5.71	92.2
4 mL L^−1^	76.1	12.7 *	25.37	6.31	5.92	92.4
2 mL L^−1^	77.2 *	11.9 *	27.17	5.42	6.15	91.9
1 mL L^−1^	76.8	12.0 *	25.14	5.23	5.54	92.3

**Table 3 plants-12-00843-t003:** Gene name and primer sequences of the genes selected for RT-qPCR validation. The six DEGs selected for validation are listed and divided according to the hormonal cluster they belong to. Each gene description is also provided. Primer forward (PF) and reverse (PR) are given for each gene.

Gene Cluster	Gene Name	Gene Description	Primer Sequence (5′-3′)
AUXIN-related genes	*YUCCA6*	Indole-3-pyruvate monooxygenase	PF: GGAGGCGGCAGTGACAAC PR: GTCGCCACCACCAACCA
	*ARF19*	Auxin response factor 19	PF: ACTTTACCTGGCTCCACAGCTT PR: TCCTAGTTGACGGGATAGATCAGAA
	*NH23*	Nudix hydrolase 23, chloroplastic	PF: CCGTTTTAGACCGTTCCGAAT PR: GAAGAAGAGGAAGCACTTAAATTTGAG
ABA-related genes	*PYL4*	Abscisic acid receptor	PF: TGAAACCCTCGTTAGCTCATGA PR: TGGAGATGGGCAGCAGAGA
	*MYB30*	Transcription factor MYB30-like	PF: GCGCGGCCCTTGAAA PR: ACCCCTGAACAAGCCTCTGA
	*PP2C62*	Probable protein phosphatase 2C 62	PF: AATTCGGAGATGCAGGTGAAA PR: TCTCTCTCCAATTCTGCTTCATTTT

## Data Availability

The data presented in this study are available on request from the corresponding author.

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
