# Peer review of "Brown Seaweed Extract (BSE) Application Influences Auxin- and ABA-Related Gene Expression, Root Development, and Sugar Yield in Beta vulgaris L."

_plants, 2023, doi:10.3390/plants12040843_

Round 1
Reviewer 1 Report
First of all, iwant to thank all the authors fro this intersting paper.
I have some question as follow:
the introduction already confirm the hypothesis of the paper befor u start ur study regarding the triggering action of the brown alage to the Auxins and ABA. I ssuggest to reduce this part or move to disscusion.
In the material and methode, you make the experiemnt in field, in pot and hydroponic with differant concentration of the sea weeds extract. In each u measure differant parameter. RNAseqfor the field , pot for RT Pcr and so on, Why u do only the field and measure all in it, instead of this splitting that sure will show sligh differance between each experiment (age, temp. humidity, ect...).
Figure 2, and Figure 3, represent a valuable data as an average of eight replicas however, there is no statistical evaluation for the differences between groups or specific comparisons using Duncan's Multiple Range Test for further comparisons between groups, N. B. Duncan's Multiple Range Test can be represented in terms of letters where, means followed by different letters are significantly different.
Error in referance entry style in line 191
Other question related the usage of seaweeds as a biotimulant or fertilizer. to collect seaweeds from sea to use as a fertilizer is againist the biodiversity and may put the seaweeds in dangers.
Author Response
February 3, 2023
Dear Editor,
I would like to resubmit the revised manuscript (plants-2179777) entitled “Brown seaweed extract (BSE) application influences auxin and ABA related gene expression, root development, and sugar yield in Beta vulgaris L.”
The manuscript has been revised following the very helpful provided comments. We want to thank you for the opportunity to re-submit our manuscript. We hope that the changes made to the manuscript were able to highlight the importance of the findings and make the current version suitable for the standard of Plants.
Looking forward to your comments.
Prof. Piergiorgio Stevanato

Reviewer 2 Report
Authors present an intersting study analysing the influence of brown seaweed extract (BSE) on differentially expression of genes in sugar beet. Some selected genes associated with auxin and abscisic acid response were tested by RT-PCR. Also phenotypic effect of BSE on plant root growth parameters as well as sugar and mineral salts yield was studied. Article is well written . Research is well planned and performed, obtained results suport conclusions. Only minor comments should be addressed before publication.
Paragraph 2.2.2. Authors stated that the PP2C62 is a part of ABA pathway. However in Table nr 3, the three genes (PYL4, MYB30 and PP2C62) are ABA-related. Also in Fig 3 as much a three genes are ABA-dependent. Correct text in line 128 by adding two more ABA-related genes, to make totally three of them, as in other parts of text or Fig. 3.
Lines 191-203 and 345; write names of genes and plant latin names in italics.
Lines 325-330; provide references as numbers in square brackets, remove names of Authors in brackets.
Paragraph 4.4 Provide details of PCR reaction. Provide from which species come from reference genes-is it also sugar beet or other plant ?
Provide details of statistical methods used in the research as for example separate paragraph.
Author Response

(The authors gave the same response as above.)

Round 2
Reviewer 1 Report
ok
accept